# DOES THE DEFINITION OF DIFFICULTY MATTER? SCORING FUNCTIONS AND THEIR ROLE FOR CURRICULUM LEARNING

## ABSTRACT

Curriculum learning (CL) relies on the simple and intuitive assumption that the non-uniform sampling of training instances based on some measure of *sample difficulty* is beneficial for learning, with the postulated benefits being faster convergence and improved test-set performance. The motivation for CL is oftentimes grounded on anthropomorphisation – humans, it is argued, often rely on curricula for their learning. However, this simple premise hinges on the notion of *sample difficulty* for which there is no established definition. Previous research on the benefits of CL begins by settling on a specific definition of difficulty, without questioning the potential bias that this *a priori* definition introduces. In the present contribution, we conduct an extensive experimental study on the robustness and similarity of the most common scoring functions for sample difficulty estimation on two benchmark datasets from the vision and audio domains. We report a strong dependence of scoring functions on the training hyperparameters, including randomness, which can partly be mitigated through ensemble scoring. While we do not find a general advantage of CL over uniform sampling, we observe that the ordering in which data is presented for CL-based training plays an important role in model performance. Furthermore, we find that the robustness of scoring functions across random seeds positively correlates with CL performance. Finally, we uncover that models trained with different CL strategies complement each other by boosting predictive power through late fusion, likely due to differences in the learnt concepts. Alongside our findings, we release a toolkit implementing sample difficulty and CL-based training in a modular fashion[1].

## 1 INTRODUCTION

As for many other concepts in *machine learning* (ML), *curriculum learning* (CL) owes part of its allure to its inspiration from human learning: in the same way that children are first taught clear and distinguishable concepts before more difficult or nuanced ones Basu & Christensen (2013); Khan et al. (2011) – the usual argument goes – ML should also benefit from a non-randomised curriculum, in which models are first confronted with easy examples before difficulty is steadily increased. In the context of differentiable functions optimised by gradient-based methods, like *deep neural network*s (DNNs), optimising on 'easier' samples in the first stages of learning is postulated to guide models towards smoother regions of the loss landscape, allowing models to quickly reach favourable parameter regions and finally converge therein Bengio et al. (2009); Wang et al. (2022). Accordingly, most common implementations of *curriculum learning* (CL) adhere to the following blueprint Hacohen & Weinshall (2019): First, all samples of a given training dataset are sorted by difficulty. Then, during training, the model is gradually exposed to more and more samples of increasing difficulty, beginning with an initial 'easy' subset, with the full dataset incrementally introduced in later stages Bengio et al. (2009); Wang et al. (2022).

Despite the intuitive motivation behind CL, defining or determining a measure of *sample difficulty* (SD) to create such a difficulty ordering remains a key open challenge Wang et al. (2022); Soviany et al. (2022); Liu et al. (2023). This aspect is crucial given that any investigations into the potential

---

[1]Link to be added after acceptance

benefits of CL depend on a well-founded quantification of SD. Importantly, a universally applicable human-inspired difficulty understanding is still missing despite efforts in this direction Mayo et al. (2023); Wang et al. (2022). Moreover, a straightforward transfer of human difficulty to CL is debatable due to significant differences in learning between humans and machines Song et al. (2024). For this reason, many model-based approaches to SD estimation have been suggested in recent years, which essentially rely on the training dynamics of a specific model-sampling combination. Yet, despite the fact that these approximations depend on training setups (e. g., model architecture, hyperparameters, or even random effects due to initialisation), there is no comprehensive study quantifying their impact. This is of particular importance given the recent trend of additionally using SD to garner insights into the structure of datasets Meding et al. (2022). We aim to overcome this limitation by exploring various configuration settings that shed more insight into the impact of these different factors.

While evidence of the benefits of CL has been reported "en masse" Hacohen & Weinshall (2019); Kesgin & Amasyali (2022); Lotfian & Busso (2019); Mallol-Ragolta et al. (2020), the methodology has not found its way into the standard training procedures. It is not clear whether this is due to the additional computational overhead and implementation complexity or an overestimation of benefits due to the factors mentioned above. A promising attempt to analyse contexts in which CL proves beneficial is provided by Wu et al. (2021). The study compares various *scoring function*s (SFs) for the estimation of SD, *pacing function*s (PFs) for the scheduling of samples during training, and different difficulty orderings (e. g., easy-to-hard, hard-to-easy, and random), providing evidence that the benefits of CL are mainly limited to shorter training time or mitigating label noise. However, the authors focus on a coarse statistical view of the problem, leaving out a more granular analysis of the individual aspects of SFs and their interplay with CL settings. Moreover, with an isolated performance analysis of the trained models, it remains an open question as to how training dynamics influence the concepts learnt by the final model states through the use of CL strategies – a question we additionally explore in this work.

In our study, we also try to overcome a common limitation in the selection of datasets for large-scale CL studies (and fundamental ML studies in general), which heavily focus on standard *computer vision* (CV) benchmark datasets Soviany et al. (2022); Li et al. (2018). To sustain comparability to existing literature, we perform our experiments on one of the most popular CV datasets, CIFAR-10. We further extend them to a popular *computer audition* (CA) task of *acoustic scene classification* (ASC) to potentially unravel modality-specific differences, as suggested in prior studies Triantafyllopoulos & Schuller (2021); Milling et al. (2024).

In summary, we aim to present a thorough analysis of properties and similarities for an extensive subset of popular SFs for SD estimation. We further evaluate the interplay of difficulty orderings with CL and analyse potential synergies between model states trained with different CL strategies. In the process, we try to bring new insights to the advantages, the limitations, and the general workings of curriculum-based learning paradigms by, at least partially, answering the following questions:

- How sensitive are different SFs to varying model architectures, training settings, as well as random seeds, and what are their limitations in a practical setting?
- To what extent do different SFs share a similar notion of SD?
- Are less training setting-sensitive SFs advantageous for achieving higher performance?
- How do different CL strategies impact model performance?
- Do models trained with varying CL strategies learn different concepts that can be combined to improve their predictive performance?

## 2 CURRICULUM LEARNING

We begin with a definition of CL followed by a definition of the most common SD methods used in our work.

### 2.1 DEFINITION OF CURRICULUM LEARNING

On a high level, CL introduces a *non-uniform sampling* over training set instances. Concretely, given a training set $\mathcal{S} = \{\{\boldsymbol{x}^i, y^i\}, i \in [0, N)\}$ of cardinality $N$ and comprising pairs of features $\boldsymbol{x}$ and

labels $y$, a hypothesis $h$ from a hypothesis class $\mathcal{H}$, and a *risk function* $L$

$$L_{\mathcal{S}}(h) := \frac{1}{N} \sum_{i=0}^{N} l(h(\boldsymbol{x}^i, y^i)), \qquad (1)$$

where $l$ is an error function representing the risk of an individual data sample, the standard *empirical risk minimisation* (ERM) rule for choosing a hypothesis $h_{\mathcal{S}} \in \mathcal{H}$ is

$$\tilde{h}_{\mathcal{S}} := ERM_{\mathcal{H}}(\mathcal{S}) \in \underset{h \in \mathcal{H}}{\arg\min}\, L_{\mathcal{S}}(h). \qquad (2)$$

If $\mathcal{H}$ is further constrained to only contain differentiable functions $h(\boldsymbol{\theta})$, and the risk function $l$ is also differentiable, then the search for the hypothesis with the lowest error can be attempted with gradient descent, where the parameters $\boldsymbol{\theta}$ of $h$ are iteratively updated towards the direction of the negative gradient. The update step for gradient descent is

$$\boldsymbol{\theta}^{t+1} := \boldsymbol{\theta}^t - \eta \nabla L_{\mathcal{S}}(h(\boldsymbol{\theta}^t)). \qquad (3)$$

Note that we explicitly avoid discussing convergence guarantees to the hypothesis class $\mathcal{H}$, such as convexity, smoothness, or Lipschitz bounds, as these are not necessary for our analysis of the sampling.

In practice, optimisation on modern DNNs is done by following *stochastic gradient descent* (SGD), where the update step is based on a randomly sampled subset (or batch) $\hat{\mathcal{S}} \subseteq \mathcal{S}$ with $1 \le |\hat{\mathcal{S}}| \le |\mathcal{S}|$. In the standard formulation of SGD, the sampling of $\hat{\mathcal{S}}$ follows a *uniform* distribution, i.e., each sample $i$ is selected according to $i \sim \mathcal{U}$ and the probability of selecting each instance is $p(i) = \frac{1}{N}$. CL imposes a different sampling distribution $\mathcal{T}$ where $p(i) \ne \frac{1}{N}$, and instead some instances receive a higher probability than others, which is changing over time. Optionally, the initial selection probability for some instances is 0.

Note also that the training of DNNs is oftentimes organised in *epochs*, where, in each epoch, instances are sampled *without replacement* until the set of training instances $\mathcal{S}$ is exhausted. In this case, CL imposes a specific *ordering* of training set instances, $\mathcal{C}^e$, for each epoch $e$, where

$$\mathcal{C}^e = \{\{\boldsymbol{x}^j, y^j\} : j \in [0, M^e), M^e \le N\} \subseteq \mathcal{S}. \qquad (4)$$

## 2.2 DEFINITIONS OF SAMPLE DIFFICULTY

There are many ways to impose a curriculum. Henceforth, we will only consider curricula which have been introduced in prior work and rely on some definition of *sample difficulty* (SD). All of them are *model-based* definitions of SD, as they rely on the existence of some trained model – usually a DNN. Based on the motivation of the original authors, we categorise them in two categories: *uncertainty-* and *geometry*-inspired. In the following, we lay out the fundamental concepts behind the respective SFs. Details relevant for implementation are further discussed in Appendix A.1.

*Uncertainty-inspired* curricula define SD based on how certain a trained DNN is about its prediction for individual instances. The simplest approach is to use the loss (**CELoss**) of each training sample given a model trained on the entire training set $\mathcal{S}$ – usually, until convergence. Alternatively, statistics derived from the training dynamics of a model trained on $\mathcal{S}$, such as the ***cumulative accuracy*** (**CumAcc**), i.e., the sum of correct classifications over all epochs divided by the total number of epochs, or the ***first iteration*** (**FIT**), which is defined as the ratio of the first epoch in which a sample is correctly classified and remains correct thereafter, and the total number of epochs. Thus, uncertainty-inspired curricula use the *confidence* (or lack thereof) of a model on each training set instance as a proxy for its difficulty.

*Geometry-inspired* curricula focus, instead, on *separability*. The consistency score (**C-score** introduced by Jiang et al. (2021) aims to measure how consistently a sample is classified correctly across different training subsets of varying sizes, with the sample in question always excluded from the training process. The underlying assumption is that samples which can consistently be classified with a small amount of training data likely require less data complexity, are more representative of their class, and are, therefore, easier to classify. We derived less computationally expensive variant, ***cross-validation loss*** (**CVLoss**). Similarly, ***transfer teacher*** (**TT**) Weinshall et al. (2018) estimates SD by analysing the classification boundaries of samples in some representation space

of high expressivity by relying on learnt embeddings from a (foundation) model pre-trained on a larger dataset, which are then classified via a *support vector classifier* (SVC). The margin between each sample and its classification boundary indicates the estimated SD. Finally, ***prediction depth* (PD)** considers representations across different layers of a trained model, based on the assumption that easy examples can be separated using shallow features, while harder examples require deeper representations. This is achieved by placing *k-nearest neighbours* (KNN) *probes* at different layers and classifying each training set example.

### 2.3 PACING FUNCTIONS

As mentioned, each epoch-based CL experiment fixes an ordering $\mathcal{C}^e$ (e. g., based on SD) for each epoch. Usually, the cardinality of $\mathcal{C}^e$, $M^e$, is iteratively increased until it encompasses all training instances. In practice, CL is abandoned after a number of epochs $E^C$ and the model is further trained using uniform sampling over the entire training set $\mathcal{S}$ for the remaining epochs. $M^e$ depends on the choice of the PF, with some PFs adding samples more slowly at the beginning and faster towards the end of training, while others start by adding samples quickly before slowing down as they approach $M^e \rightarrow N$.

## 3 RESULTS & DISCUSSION

This section presents our core findings. We first begin by a comparison of SFs in Section 3.1. Following that, we continue with a discussion of trade-offs involved in CL in Section 3.2. Details on the investigated datasets, CIFAR-10 and DCASE2020, are discussed in Appendix A.2 and the *convolutional neural network* (CNN)-based model architectures are presented in Appendix A.3.

### 3.1 SCORING FUNCTIONS

#### 3.1.1 ROBUSTNESS TO TRAINING HYPERPARAMETERS

We begin by considering the *robustness* of SFs with respect to the random seed, model architecture (& initialisation), and the combination of optimiser and learning rate. We investigate these by training different experimental setups, changing only one setting – seed, model, or optimiser-learning rate combination – at a time w. r. t. our baseline configuration. Overall, we choose six different variations per training setting and obtain one SD ordering per SF and variation.

Table 1 summarises the robustness of different SFs per training setting for CIFAR-10 and DCASE2020. We computed the mean and standard deviation of pairwise Spearman rank correlations of the difficulty orderings obtained for each independently varied training setting. The mean pairwise correlation is, in most cases, moderate ($\rho \geq 0.4$) or strong ($\rho \geq 0.6$). However, there are notable differences across different SFs. Broadly, real-valued SFs, i. e., *cross-entropy-loss* (CELoss) and CVLoss show lower agreement than discrete-valued ones. Rather than an inherent limitation, we see this is as a corollary of the specificity of each SF. Discrete-valued SFs have a low number of unique values; for CumAcc and FIT it is the number of epochs, whereas for PD it is the number of layers. This necessitates a *tie-breaking* procedure. In our case, rather than a random ordering in case of examples with identical difficulty, we chose to adhere to their original ordering in the dataset (e. g., the alphanumeric ordering of the file name), which resulted in artificially boosting the agreement for these SFs over others. For a more detailed discussion, see Appendix A.4.

Results are consistent across datasets but do vary with respect to the type of training setting. Changing the random seed results in overall higher agreement, with a change in the model architecture exhibiting the lowest agreement. This shows how different models – with different inductive biases – encapsulate a different notion of SD, with other training settings such as the optimiser and learning rate adding further variability. Nevertheless, even changing the random seed alone – and keeping all other things equal – results in a low to moderate amount of disagreement. As the random seed controls both the model initialisation and the order in which examples are presented during training, it is evident that these two aspects of training play a non-negligible role when discussing SD.

In summary, we conclude that an SD ordering is context-dependent, as it is influenced by the choice of model, training hyperparameters, and even random seed selection. The corollary is that the

Table 1: Training setting robustness in terms of agreement of difficulty rankings obtained with varying training settings. The respective training setting of each column is varied with respect to the reference configuration. We then report the mean and standard deviation of Spearman correlation coefficients ($\rho$) computed between unique pairs of training setting variations.

| SF | Seed | Model | Optim + LR |
| --- | --- | --- | --- |
| **CIFAR-10** | | | |
| CELoss | .507 ± .026 | .428 ± .043 | .483 ± .055 |
| CVLoss | .676 ± .007 | .629 ± .045 | .689 ± .022 |
| CumAcc | .760 ± .008 | .557 ± .101 | .752 ± .019 |
| FIT | .586 ± .033 | .416 ± .076 | .623 ± .019 |
| PD | .790 ± .012 | .653 ± .076 | .799 ± .032 |
| TT | – | .648 ± .025 | – |
| **DCASE2020** | | | |
| CELoss | .410 ± .060 | .415 ± .115 | .369 ± .041 |
| CVLoss | .591 ± .018 | .579 ± .044 | .556 ± .046 |
| CumAcc | .821 ± .012 | .590 ± .099 | .758 ± .048 |
| FIT | .604 ± .020 | .475 ± .084 | .513 ± .052 |
| PD | .748 ± .023 | .694 ± .046 | .683 ± .068 |
| TT | – | .523 ± .191 | – |

model-based curricula we investigate later can be seen as 'global' curricula in only a relative sense, as each training setting results in a different SD ordering.

### 3.1.2 ENSEMBLING SCORING FUNCTIONS

The strong influence of randomness across all SFs observed in Section 3.1.1 poses concerns towards the legitimacy of model-based SD prediction. Following Wu et al. (2021) and Kesgin & Amasyali (2022), we consider *ensembling of difficulty orderings* as a simple mitigation strategy. The underlying hypothesis of this approach is that the effects of random variations can be counteracted by averaging over several difficulty predictions obtained from different random seeds for each sample, thus unravelling a more robust and less biased SD estimation. In order to investigate this hypothesis, we first train a total of nine new models per dataset by varying the random seed of the EfficientNet-B0 reference configuration. Per dataset and SF, we obtain 15 difficulty estimations, the settings of which only vary in the underlying random seed. We omit CVLoss due to computational costs and TT due to a lack of dependence on randomness from the discussion. From the 15 difficulty estimations, we form three ensemble difficulty orderings for each of the varying ensemble sizes {1,2,3,4,5}, as described in Appendix A.5, while ensuring no overlap in the underlying SFs across the ensembles.

Fig. 1 illustrates the average pairwise correlation across the three considered ensemble difficulty orderings for each ensemble size. It is apparent that the pairwise correlation consistently increases with larger ensemble sizes across datasets and SFs. SD orders obtained from the same experimental settings but different random seeds thus agree more with each other if the orders are built via an ensemble over multiple random seeds. This result is in line with our hypothesis, clearly indicating that difficulty orderings become more robust towards randomness with increasing ensemble sizes.

### 3.1.3 AGREEMENT ACROSS SCORING FUNCTIONS

Given the previous discussion on the characteristics of SFs on an individual level, we now extend our analysis to the similarity of difficulty notions *across* SFs. For this, we calculate pairwise Spearman rank correlation coefficients of the different SFs based on their ensembles w. r. t. seed, model architecture (including initialisation), and optimisation routine. The underlying hyperparameter grid for this analysis is further specified in Table 5 in Appendix A.6. Table 1 summarises the SF agreement per ensemble type as an average of the pairwise agreements with detailed results presented in Fig. 4 in Appendix A.7.

Evidently, almost all SFs share a very similar notion of difficulty. Interestingly, the agreement across SFs is even higher than most agreements within the same SF with different random seeds. While this might seem surprising at first, the reasons for this may lie in the fact that most approaches are based on identical model trainings. For instance, a likely scenario might be that a model is confronted with a certain sample early on during training and is able to learn it quickly. Consequentially, the sample

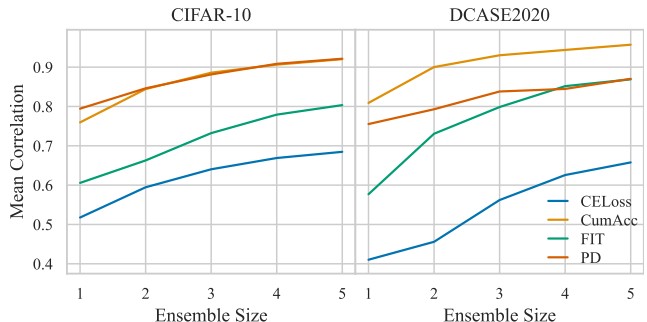

Figure 1: Correlation of SFs with increased ensemble sizes for the datasets CIFAR-10 and DCASE2020. The ensemble size encapsulates how many individual orderings – obtained from different random seeds – are considered to build one ensemble. For each SF, we report pairwise Spearman correlations across three ensemble orderings with the same ensemble size.

Table 2: Mean SF agreement for CIFAR-10 and DCASE2020. Reported is the mean pairwise agreement of ensemble SFs under variation of either random seeds, model architectures, or optimiser-learning rate combinations.

| Variation | CIFAR-10 | DCASE2020 |
|---|---|---|
| Seed | $.724 \pm .097$ | $.689 \pm .129$ |
| Model | $.673 \pm .085$ | $.652 \pm .132$ |
| Optim + LR | $.732 \pm .112$ | $.724 \pm .080$ |

might be correctly classified in all epochs, making it both 'easy' in the context of CumAcc and FIT, and having a persistently low loss value, making it 'easy' in the context of CELoss. Only PD shows a low, yet still moderate correlation to all other SFs of above .5 in almost all cases. With all other approaches agreeing to more than 70 % in all but one case, their ensemble-based difficulty orderings generally seem to be good estimators for model-based SD in the context of CL.

## 3.2 CURRICULUM LEARNING

So far, we only investigated SFs in isolation regarding their alignment of difficulty orderings. This section focuses on the practical implications of SD orderings for CL. We perform experiments as described in Section 2.3, based on the ensemble SD orderings obtained from Section 3.1.3.

### 3.2.1 BENEFITS OF ROBUST SCORING FUNCTIONS FOR CURRICULUM LEARNING

Despite the many nuanced aspects of CL, it is well documented that the order, in which data is presented to the DNN at training time is of high importance with a hard-to-easy (*anti-curriculum learning* (ACL)) ordering being generally less preferable compared to the common easy-to-hard (CL) ordering Wu et al. (2021). We verify this claim in Appendix A.8 showing that performance consistently worsens when changing the order from CL to *random curriculum learning* (RCL) to ACL with effects being more prominent the slower new examples are added to the training. Beyond the apparent importance of data orderings in these extreme cases, we further aim to investigate the more subtle impacts SF-based orderings have on CL in the following. For this purpose, we revisit the robustness experiments of SFs from Section 3.1.2. We hypothesise that ensemble SFs of larger size – being more robust to randomness – should have a clearer notion of difficulty, which in turn should lead to a more reliable difficulty ordering and thus have benefits in a CL setting.

In order to investigate this hypothesis, we base our CL experiments on the ensemble orderings previously investigated in Section 3.1.2. We use three different orderings per ensemble size and three different random seeds, resulting in overall 60 curriculum experiments per SF for each dataset. Figure 2 displays the average pairwise correlation for each ensemble size, i. e., the values on the $y$-axis in Figure 1, versus the average performance of the CL-based training with the same ensemble size.

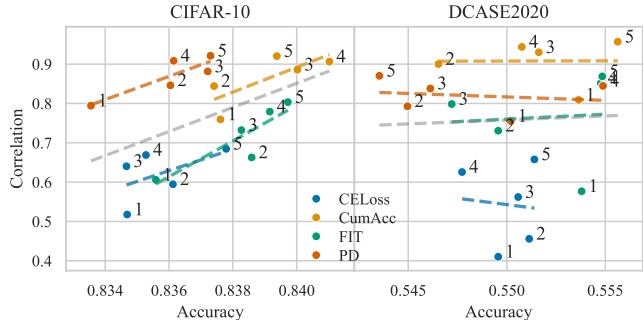

Figure 2: Comparison of SF robustness and CL performance for CIFAR-10 and DCASE2020. For each scoring function, we evaluate ensemble orderings of different ensemble sizes, noted as a number next to each point. The $y$-axis represents the pairwise correlation across the ensembles of the respective ensemble size (cf. $x$-axis in Figure 1) as an indicator of SF robustness. The $x$-axis displays the average accuracy of CL experiments based on the corresponding ensemble orderings. Coloured and grey dashed lines are linear least-squared-error fits per SF and across all SFs, respectively. The slope of the lines indicates whether trends of higher SF robustness and higher CL performance exist.

At this point, results for CIFAR-10 and DCASE2020 diverge. The outcomes for CIFAR-10 clearly support our hypothesis both within (correlation across same-coloured points) and across (correlation across all points) SFs, with high *Pearson correlation coefficient*s (PCCs) between accuracy and correlation of .760 and .498, respectively. The first value is calculated as the average correlation for each SF separately (macro PCC), while the second value is based on the correlation of all points independent of the concrete SFs (micro PCC). This aligns with our expectation that a more robust SD ordering will result in improved performance when used as the starting point for CL.

DCASE2020, however, adds more ambiguity to the discussion: both the average correlation within and across SFs show close to no correlation of -0.047 and 0.045, respectively. In this case, a higher agreement on the SD did not translate to improved performance when this ordering was used for CL, indicating that other factors might influence CL behaviour. While we find clear evidence supporting the benefit of more robust SFs for CL on the CIFAR-10 dataset, no such evidence can be reported for DCASE2020, suggesting further investigation into how SF robustness and other factors contribute to CL performance.

### 3.2.2 SCORING FUNCTION AND PERFORMANCE

Having compared the effects of different orderings within the CL setting, the next aspect we aim to investigate is whether the CL methodology is in any way superior to standard *deep learning* (DL) training beyond the limited training time scenario reported in Wu et al. (2021). To test this, we select the 264 CL experiments for the CIFAR-10 and 216 for the DCASE2020 dataset. We report the best-performing combination for easy-to-hard (CL), hard-to-easy (ACL), and the random ordering (RCL) averaged across three seeds, respectively. For comparison, we use the 15 models from 3.1.2, each replicating the EfficientNet-B0 reference configuration across different random seeds, as baselines. We report the performance of the single best model (B1), the average performance of the best three models (B3), the best five models (B5), and all baselines (B15). This setup allows us to investigate whether CL, RCL, or ACL can offer performance benefits over standard training across both datasets in order to challenge the findings of Wu et al. (2021), which reported no improvement over the baseline.

Table 3 provides an overview of the performance experiments. Results are inconclusive. For CIFAR-10, the best performance is indeed achieved by a CL-based configuration using the computation-heavy C-score ordering. However, the next best performances are achieved by all baselines, followed by RCL and, finally, ACL showing the worst performance. For DCASE2020, the best CL setting cannot reach the performance of the best baseline run but outperforms the average performance from the best 3 to 15 baselines, with also ACL and RCL outperforming the average overall baseline runs. A clear plug-and-play improvement of CL over standard training across datasets can therefore not be concluded, which falls in line with the results of Wu et al. (2021).

Table 3: Performance comparison between the averages of the best standard training baselines (abbreviated with B) and the best training settings for CL, RCL, and ACL averaged across three seeds.

| CIFAR-10 | | | DCASE2020 | | |
|---|---|---|---|---|---|
| **SF** | **Type** | **Accuracy** | **SF** | **Type** | **Accuracy** |
| C-score | CL | .844 | – | B1 | .583 |
| – | B1 | .839 | TT | CL | .577 |
| – | B3 | .839 | – | B3 | .576 |
| – | B5 | .838 | – | B5 | .571 |
| – | B15 | .834 | CELoss | ACL | .560 |
| Random | RCL | .829 | Random | RCL | .558 |
| CELoss | ACL | .829 | – | B15 | .555 |

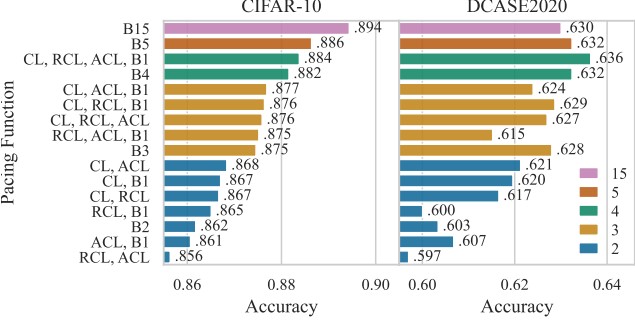

Figure 3: Late fusion results for combinations of curriculum (CL), random curriculum (RCL), and anti-curriculum learning (ACL), as well as the best baselines abbreviated with B; for instance, B4 represents the fusion of the best 4 baseline runs.

### 3.2.3 LATE FUSION PERFORMANCE

Our last line of experiments focuses on the question of whether models trained with a focus on different samples arrive at states that are complementary to each other. We investigate this through the lens of *late fusion*, which is performed by averaging predictions. Therefore, we select the best-performing model trained with easy-to-hard, hard-to-easy, and random orderings alongside the baselines from 3.2.2. We denote the fusion of the top two baselines as B2, the fusion of the top three baselines as B3, and so forth.

Figure 3 provides an overview of the performance obtained through late fusion of differently trained model states. Across all experiments, we notice a clear performance increase when fusing multiple models, as would be expected from the literature Dietterich (2000); Hu et al. (2018). For CIFAR-10, the highest performance across all experiments is achieved by the fusion of all baseline models. In contrast, for the DCASE2020 dataset, the highest performance is obtained from the fusion of CL, ACL, RCL, and the best baseline. Among fusions of two, three, and four models, we find that combinations including any of the CL-based orderings tend to outperform baseline fusions of the same size across both datasets, especially if CL and ACL are part of the combination. This is particularly interesting as ACL and RCL have a lower performance compared to the best baselines (cf. Table 3). The fact that the fusion of CL and ACL performs better suggests that these opposing difficulty orderings may complement each other, indicating an exploitable difference in the learnt concepts between CL and ACL.

## 4 CONCLUSION

Our experiments confirm the dependence of curriculum learning on the choice of scoring and pacing functions, with additional random effects instigated by the choice of hyperparameters, models, datasets, and random seeds. Ensembling difficulty scores within a scoring function results in more robust orderings which lead to higher agreement on what constitutes a difficult example across different scoring functions. However, non-uniform sampling based on sample difficulty does not result in consistently better performance, irrespective of whether the ordering is ascending (easy-

to-hard) or descending (hard-to-easy). Similarly, training on more 'robust' orderings obtained via ensembling does not always result in performance gains either. Nevertheless, different orderings result in complementary model states that can improve their performance by model ensembling. Collectively, our findings confirm previous work which argued that curriculum learning, as currently applied, is not a panacea, but rather conditional on experimental choices. Accordingly, this challenges anthropomorphising assumptions underpinning the colloquial motivation of curriculum learning and calls for more theoretical investigations of curricula in the context of DNNs – which this work aims to inspire.

## 5 BROADER IMPACT

In the era of big data and *scaling laws*, reducing the computational footprint of experiments can 'level the game field' for groups without access to the computational resources of large industrial labs while reducing the environmental cost of DNN experiments; thus, the obvious appeal of CL. Beyond that, curriculum learning *feels right* but remains an elusive concept. Understanding when it works – and why – is important towards establishing its theoretical foundations. Our study complements previous work in that direction by contributing several experimental observations which demonstrate the need for more research in order to clearly establish CL as a training paradigm that reigns superior over standard ERM training.

## 6 RELATED WORK

Experimental studies surrounding the potential benefits of CL-based learning paradigms are naturally limited through computational resource demands to specific contexts and settings, with the most extensive examples being restricted to CV datasets. Consequentially, contradicting evidence is reported in some cases across different works, leaving the question of the advantages of CL open.

For instance, Hacohen & Weinshall (2019) reported a higher performance for models trained with a curriculum, but the advantages of CL were higher for more difficult datasets and the impact of CL was concentrated at the earlier stages of the training. On the other hand, the most extensive CV-based study with respect to the number of trained models Wu et al. (2021) is purely based on a set of model-based SFs. It considers ensembles to achieve higher stability in the difficulty orderings. It agrees with the conclusion that CL shows faster learning speed at the beginning of training but cannot find evidence for a significantly higher performance compared to the baseline. As a random SD ordering is reported to yield similar performance benefits as any order obtained from the applied SFs, it is hypothesised that the advantages of CL can be attributed primarily to the dynamic dataset size during training. In particular, PFs tend to perform better when they quickly incorporate more difficult samples and saturate on the full dataset.

Beyond the realm of CV, Lotfian & Busso (2019) performed CL experiments for *speech emotion recognition* (SER). The study considers both model-based SFs obtained from the models' loss functions as well as human-based SFs derived from the inter-rater agreement. The authors report marginally higher performance for the model-based ordering and noticeably higher performance for the rater-based ordering, compared to the baseline. Contrary to Wu et al. (2021), they do not find any comparable benefits from random orderings. A study on CL for *automatic speech recognition* (ASR) was performed by Karakasidis et al. (2024), which considers difficulty-orderings based on the utterance length, a model-based SF, and *self-paced learning* (SPL). The length of the utterance, which aims to incorporate human priors into the SD estimation, achieved the lowest performance overall. Contrary to Hacohen & Weinshall (2019), the best results are obtained with SPL, yet CL-based approaches still performed on par. It is further reported that the benefits of employing PFs are particularly eminent in the later stages of the training, contradicting both Hacohen & Weinshall (2019) and Wu et al. (2021).

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

## A APPENDIX

### A.1 IMPLEMENTATION DETAILS OF SCORING FUNCTIONS

**Consistency Score (C-score)**: Jiang et al. Jiang et al. (2021) draw 2000 subsets for each subset ratio $s \in \{10\%, \ldots, 90\%\}$, creating publicly available proxies of the C-score[2] for the CIFAR-10 dataset based on 18 000 models. Given the high computational costs associated with the calculation of the C-score, we limit our analysis to the publicly available C-scores for CIFAR-10 and do not determine them for the ASC task.

**Cross-Validation Loss (CVLoss)**: As a computationally cheaper – yet still expensive alternative to the C-score, we draw inspiration from the *cross-validated self-thought* (CVST) SF Wu et al. (2021) and a loss-based C-score variant Kesgin & Amasyali (2022), defining the CVLoss as the average loss of a sample when it is held out in a randomly partitioned $k$-fold cross-validation setting. The loss-based version allows for a more fine-grained SD estimation compared to the accuracy-based C-score but does not account for the effects of differently sized training subsets. In our experiments, we use $k = 3$ partitions, allowing us to obtain one score for each sample from the training of three models. This approach achieves a reasonable trade-off between computational cost and the model's expected performance given the size of the training subset.

**Prediction Depth (PD)**: Another combination of DNN-based representations and traditional machine learning algorithms for classification is the PD introduced in Baldock et al. (2021). PD considers the representations extracted from the final state of a trained model at different layers, which supposedly represent different views of the same data sample in the form of low- to high-level features. The representations extracted from the layers are then classified by KNN probes with $k = 30$. Baldock et al. Baldock et al. (2021) place probes both at the input of a model and after applying the Softmax function to the output. For *residual network* (ResNet) architectures, probes are positioned after the normalisation layer in the stem and following the skip connection in each residual block. For VGG architectures, probes are placed after each convolutional layer. We extend the ResNet probe placement strategy to the EfficientNet family by inserting probes after each block. As both CNN10 and CNN14

---

[2]https://pluskid.github.io/structural-regularity/

are inspired by the VGG architecture, we place probes after each normalisation layer and the first linear layer. This results in 19 probes for ResNet50, 20 for EfficientNet-B0, 36 for EfficientNet-B4, 12 for CNN10, and 16 for CNN14. PD is defined as the depth of the first layer in which the KNN probe's prediction matches the model's final prediction and aligns for all subsequent probes in deeper layers. Easier samples are expected to be separable at lower layers, while harder samples should require higher-level features for classification. While Baldock et al. Baldock et al. (2021) do not assign a SD score if the prediction of the final probe differs from the network's prediction, we assign the maximum depth to obtain a difficulty measure for the complete training subset. To ensure that the KNN probes are computationally feasible, we limit the representation size at each layer to 8192. If any representation exceeds this limit, *global average pooling* (GAP) is applied to the spatial dimensions of the input, while the number of channels is preserved. Beyond, GAP is applied to match the spatial dimensions of the input, allowing for an equal contribution of time and frequency information in the case of our ASC task. However, the use of GAP may lead to the loss of critical features, potentially challenging the SF's ability to accurately estimate the SD.

To reduce the computational costs of CVLoss, we reduce the training time to 25 epochs. Moreover, TT is only calculated over the pre-trained versions of the three model architectures, as the SF requires a pre-trained model by design, and the SVC fitting on the target task does not allow for varying random seeds or optimisation routines. Whenever the selection of a specific model state is required to calculate a SF, we choose the best-performing state analogous to Appendix A.6.

## A.2 DATASETS

Our experiments are performed on two 10-class classification datasets from the domains of CA and CV, respectively. Both datasets vary in size, comprise samples with different representation sizes, and have different baseline accuracies.

**CIFAR-10**: The subset of the tiny image recognition dataset referred to as CIFAR-10 Krizhevsky et al. (2009) is one of the most popular benchmark datasets for understanding DL training in general and for CV in particular Glorot & Bengio (2010); Zhang et al. (2017). It comprises 60 000 images, divided into 50 000 samples for training and 10 000 for testing, with both subsets being fully balanced across ten classes, which can coarsely be categorised as animals and vehicles. The images are characterised by a rather small representation size of $32 \times 32$ pixels with three colour channels. Modern state-of-the-art approaches are able to achieve very high performance on the dataset Tan & Le (2019; 2021); Dosovitskiy et al. (2021); Feng et al. (2022). We upsample all images using bilinear interpolation to balance accuracy and computational costs. This results in a representation size of $64 \times 64 \times 3$. It is important to note that we employ a train-test split without an explicit validation subset, as the main focus of our work is on understanding SD on the full training dataset rather than achieving state-of-the-art performance. This approach is applied consistently across both datasets, as well as the baseline and CL-based training experiments.

**DCASE2020**: For the CA dataset, we choose task 1a of the DCASE2020 challenge Heittola et al. (2020). It comprises 13 962 training and 2968 testing audio samples of 10 s length across many different cities, which are recorded with different real and simulated devices. Both the training and testing subsets are nearly class-balanced. The labels indicate the acoustic scene, representing the type of location where the audio samples were recorded. Examples are public transport vehicles, open spaces, and indoor environments. In our experiments, raw audio samples are transformed into log Mel-spectrograms with 64 Mel bins, a window size of 512 ms, and a hop size of 64 ms, following the spectrogram extraction process outlined in Kong et al. (2020). This results in a representation size of $64 \times 1001 \times 1$, where each element corresponds to the magnitude of the signal's energy in a specific Mel-frequency bin and time frame.

## A.3 NETWORK ARCHITECTURES

Given the image-like nature of both the CIFAR-10 samples and the log-Mel spectrograms in the DCASE2020 datasets, we decide to employ CNN-based architectures, which are well-established in the literature of the respective tasks Dong et al. (2021); Ding et al. (2024).

**ResNets**: ResNets He et al. (2016) are among the most impactful DNN architectures in DL, playing a significant role up to this day. The introduction of residual (or skip) connections allowed for

developing very deep CNN architectures. In our experiments, we specifically employ the ResNet50 architecture for the CIFAR-10 dataset.

**EfficientNets**: The EfficientNet family Tan & Le (2019) is inspired by residual connections of the ResNet architecture and further employs mobile inverted bottlenecks Sandler et al. (2018) as well as squeeze-and-excitation blocks Hu et al. (2018). The base model, EfficientNet-B0, is optimised to balance task accuracy and computational complexity, while larger versions are derived via compound scaling of the architecture's depth, width, and resolution. In our experiments, we utilise EfficientNet-B0 for both the CIFAR-10 and DCASE2020 datasets, and the larger EfficientNet-B4 only for the CIFAR-10 dataset.

**PANNs**: The CNN10 and CNN14 architectures were introduced as part of the *large-scale pre-trained audio neural network* (PANN) Kong et al. (2020) family for spectrogram-based CA tasks. These models follow a more traditional CNN design, comprising regular convolutional and batch normalisation layers without skip connections, and are inspired by the VGG architecture Simonyan & Zisserman (2014). CNN14 can be considered a scaled-up version of the CNN10 architecture. Both models are only utilised for the DCASE2020 dataset.

Overall, we consider two types of initialisation for all presented networks: randomly initialised and pre-trained on the large datasets ImageNet Russakovsky et al. (2015) for the ResNet and EfficientNet architectures, and AudioSet Gemmeke et al. (2017) for the PANN architectures.

## A.4 Specificity of Scoring Functions

A misleading conclusion that might be drawn from Section 3.1.1 and Fig. 1 would be that higher robustness to training hyperparameters would automatically serve as a sign for a better SF. However, there are some methodological aspects which counteract this interpretation, namely, that a comparison of difficulty orderings assumes unique and distinct difficulty values for each sample, which cannot be guaranteed for some SFs.

In particular, some SFs place a natural limit on the number of unique SD values that they admit. For instance, CumAcc and FIT are limited with respect to the number of epochs used for training. Similarly, PD is limited by the number of layers in a network, whereas, in contrast, CELoss, CVLoss, and TT all take values in $\mathbb{R}^+$. By necessity, SFs with a limited range of values require a *tie-breaking* strategy – recall that we opted to use the index of the sample provided by the original dataset (e. g., the alphanumeric ordering of file names). This is necessary to avoid random orderings but has the negative effect of limiting their *specificity* in identifying the difficulty of examples and artificially boosting the agreement of these SFs.

An overview of the *granularity* of the different SFs in the single-score as well as the ensemble context is given by Table 4. Naturally, FIT, CumAcc, and PD show the coarsest granularity, as their discrete nature only allows for a limited number of difficulty predictions – at least in the single-score context.

The loss- or margin-based scoring functions CELoss, CVLoss, and TT, as well as C-score, on the other hand, have (almost) completely unique difficulty values assigned to each sample – at least in the ensemble version. Any non-uniqueness, especially in the single-score version, results from limited numerical precision for very small loss values. Therefore, they provide a clear ordering in terms of their estimation of difficulty.

## A.5 Difficulty Ordering and Ensemble Scoring

Any of the SFs introduced in Section 2.2 can assign a difficulty score to each individual sample in the training set, however, these scores are generally not normalised. For the purpose of CL, as we apply it, the exact difficulty score is not of particular importance. Instead, we solely rely on a difficulty-based ordering derived from the SD scores provided by the respective SF. In cases where multiple samples share the same exact difficulty score, they are sorted according to the original dataset ordering to avoid introducing randomness into the process. Ensembles across multiple SFs of the same type can be constructed by averaging the SD scores from each SF prior to creating the final difficulty ordering Kesgin & Amasyali (2022).

---

[3]The ensemble ordering consists of three different scores, as TT only utilizes pre-trained models.

Table 4: Granularity of CIFAR-10 and DCASE2020 SF difficulty distributions for single seed ordering and ensemble orderings. We report the number of unique difficulty values and the maximum number of samples assigned to a single difficulty value (bin).

| Ensemble Size | 1 | | 6 | |
|---|---|---|---|---|
| **Scoring Function** | **Unique** | **Max Bin** | **Unique** | **Max Bin** |
| **CIFAR-10** | | | | |
| CELoss | 17 424 | 5164 | 49 844 | 10 |
| CVLoss | 32 872 | 768 | 50 000 | 1 |
| CumAcc | 33 | 18 279 | 157 | 4033 |
| FIT | 46 | 18 279 | 34 286 | 3824 |
| PD | 20 | 21 135 | 196 | 10 915 |
| TT[3] | 50 000 | 1 | 50 000 | 1 |
| C-score | – | – | 50 000 | 1 |
| **DCASE2020** | | | | |
| CELoss | 8402 | 470 | 13 962 | 1 |
| CVLoss | 13 262 | 22 | 13 962 | 1 |
| CumAcc | 32 | 1889 | 151 | 670 |
| FIT | 48 | 1889 | 12 384 | 670 |
| PD | 21 | 4119 | 203 | 2881 |
| TT[3] | 13 962 | 1 | 13 962 | 1 |

## A.6 BASELINE MODELS

To identify the set of hyperparameters best suited to each model and dataset and establish a baseline to compare with, we run a set of preliminary experiments iterating over a small set of hyperparameters (cf. Table 5). In all cases, models are trained for the full 50 epochs, and the final model selection is based on the best-performing model state on the respective validation set. Overall, we train 54 models per dataset, and the best-performing training configuration for each model and initialisation is reported in (cf. Table 6).

Table 5: Overview of variations in random seed, model architecture (with EfficientNets abbreviated to B0 and B4) and initialisation, and optimiser-learning rate combinations used for SF calculation. The suffix -T indicates pre-training (on ImageNet for CIFAR-10 and AudioSet for DCASE2020) across model architectures. Each training setting is varied independently, with other parameters fixed to the first entry in each row, based on the EfficientNet-B0 reference configuration.

| Variations | CIFAR-10 | DCASE2020 |
|---|---|---|
| Seed | 1, 2, 3, 4, 5, 6 | 1, 2, 3, 4, 5, 6 |
| Model | B0, B0-T, B4, B4-T, ResNet50, ResNet50-T | B0, B0-T, CNN10, CNN10-T, CNN14[4], CNN14-T |
| Optim + LR | Adam, SAM, SGD (all with .001, .01) | SAM, Adam, SGD (all with .01, .001) |

Even though the best performances for CIFAR-10 and DCASE2020 are obtained with pre-trained versions of ResNet50 and CNN14, respectively, we select the best-performing EfficientNet-B0 configuration for both datasets as a baseline reference for further analysis. This choice is motivated by the model being the only architecture employed for both datasets and its comparably low parameter count, allowing for a more efficient training. Training the model from scratch further reduces the potential impact of pre-training data. The reference baseline performances are thus 0.839 for CIFAR-10 and 0.583 for DCASE2020, achieved using the Adam optimiser with a learning rate of 0.001 and the SAM optimiser with a learning rate of 0.01, respectively. Nonetheless, we consider all models to investigate the impact of varying training settings on SFs in 3.1.

## A.7 PAIRWISE AGREEMENT OF SCORING FUNCTIONS ACROSS HYPERPARAMETERS

A more detailed view on the pair-wise correlations between different scoring function ensembles is presented in Figure 4.

---

[4]The learning rate for CNN14 was reduced by a factor of ten, as the model did not converge with all other parameters fixed to the first entry in each row.

Table 6: Best baseline performance of each model on CIFAR-10 and DCASE2020 with variations in optimisers and learning rates as shown in Table 5. The suffix -T indicates pre-training (on ImageNet for CIFAR-10 and AudioSet for DCASE2020). All models were trained for 50 epochs, with final selection based on the best validation performance.

| Model | Optimiser | Learning Rate | Accuracy |
|---|---|---|---|
| **CIFAR-10** | | | |
| ResNet-50-T | SAM | .001 | .949 |
| EfficientNet-B4-T | SAM | .001 | .945 |
| EfficientNet-B0-T | SAM | .001 | .936 |
| EfficientNet-B4 | Adam | .001 | .848 |
| EfficientNet-B0 | Adam | .001 | .835 |
| ResNet-50 | Adam | .001 | .813 |
| **DCASE2020** | | | |
| CNN14-T | Adam | .0001 | .678 |
| CNN10-T | SAM | .01 | .653 |
| EfficientNet-B0-T | SAM | .01 | .644 |
| CNN10 | SAM | .001 | .609 |
| CNN14 | SAM | .001 | .595 |
| EfficientNet-B0 | SAM | .01 | .583 |

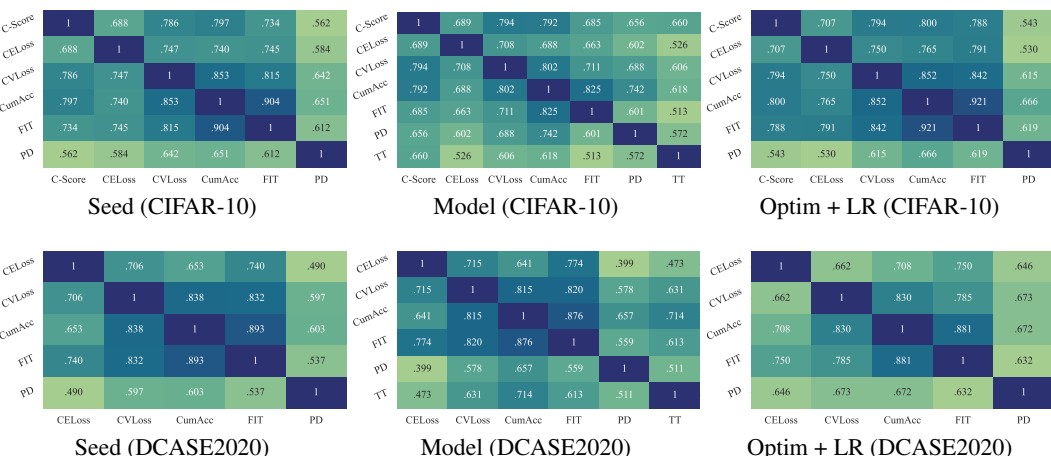

Figure 4: Agreement of different scoring functions with varying random seeds (left), models (middle) and optimiser & learning rate combinations (right). Displayed is the pairwise Spearman correlation of the respective ensemble orderings of ensemble size six.

## A.8 DIFFERENT ORDERINGS AND PACING FUNCTIONS

We evaluate how model performance is impacted when examples are presented in the intended curriculum ordering (easy-to-hard; CL), the reverse ordering (hard-to-easy; ACL), and a completely random ordering (RCL). For a robust evaluation across the different orderings, we chose the seed-based ensembles for all SFs. Moreover, to limit computational demands, we excluded the CVLoss and FIT SFs from the subsequent performance experiments, as a correlation above 80 % is noted to CumAcc for seed-based ensembles across both datasets.

For training, we selected the EfficientNet-B0 architecture in combination with the best baseline training setup, as preliminary experiments indicated that this configuration also shows high performance in combination with different pacing functions. We evaluated each SF across four PFs (logarithmic, root, linear, exponential), starting with an initial training dataset size of $b = 20\%$ and a saturation on the full dataset after $a = 50\%$ and $a = 80\%$ of the training iterations. Following the approach of Hacohen & Weinshall (2019), we incorporated new training samples in a class-balanced manner, such that the training subsets remain balanced throughout training, assuming the full dataset is balanced. Analogous to Wu et al. (2021), we also delayed the introduction of new samples until all examples in the current training subset have been used at least once. We trained each model for a

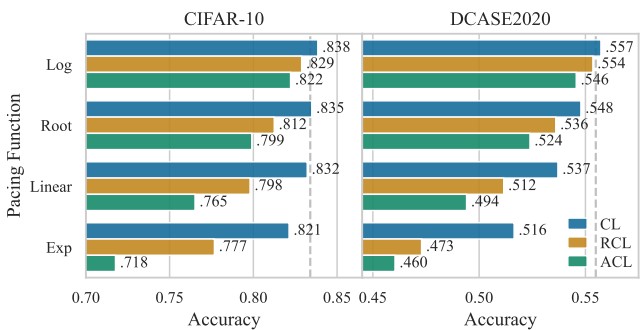

Figure 5: Mean PF performance on CIFAR-10 and DCASE2020, averaged across SFs, saturation fractions, and three seeds. Each bar represents the average performance for each PF and curriculum ordering. The grey dashed vertical lines indicate the baseline performance, averaged across the 15 random seeds.

number of steps equivalent to 50 epochs of the full dataset and replicate the experiments across three different seeds, averaging the performance to ensure robustness. Overall, this lead to a total of 264[5] CL experiments for the CIFAR-10 and 216 for the DCASE2020 dataset.

The results of our experiments – averaged over SFs – as well as the baseline performance without CL are summarised in Figure 5. For each ordering strategy, PFs are sorted from top to bottom by decreasing saturation speed, i. e., pacing functions at the top quickly introduce new samples to the training set, while those at the bottom initially maintain the number of samples close to the initial training set. Firstly, we observe a clear trend towards better performance obtained from quickly saturating PFs, which is in line with the findings in Wu et al. (2021). Only logarithmic PFs combined with a CL ordering, but not with a RCL or ACL ordering, are able to marginally surpass the baseline on both datasets.

Our experiments further show a clear performance advantage of CL over ACL, with RCL scoring in between CL and ACL. Despite being less prominent for the quickly saturating PFs, this difference steadily increases with decreasing saturation speed. In the case of the slowly saturating PFs in particular, models are trained for a long time with only the easiest (CL), the most difficult (ACL), or random samples (RCL), respectively. While most curriculum orderings and PF lead to detrimental performance to some degree, which can likely be attributed to an effect of overfitting from which the model struggles to recover at the later stages of the training, the effects are the strongest for ACL and the weakest for CL. We conclude that model training is clearly negatively impacted if confronted with difficult samples first. This finding underlines the importance of the data that models are exposed to early on. It provides evidence supporting the concepts behind CL and adds to the validity of model-based SFs for SD estimation.

## A.9 REPRODUCING OUR WORK

In the following, our experiments and corresponding discussions are structured sequentially, first focusing on the behaviours of SFs, then analysing the performance of curriculum-based training settings. All experiments are conducted using Python 3.10 and PyTorch 2.1.0 Paszke et al. (2019). For any hyperparameter not explicitly modified, the default values provided by PyTorch are used. Our work is based on two publicly-available packages[6]. They allow for obtaining SD scores from arbitrary classification datasets and supports curriculum-based training. The code to reproduce our experiments is publicly available on GitHub[7]. Training is performed across a variety of GPU architectures, including NVIDIA GeForce GTX TITAN X, GeForce RTX 3090, and A40 GPUs. Given the diverse nature of the utilised GPUs, we omit an analysis of training time comparisons between the standard training baseline and curriculum-based experiments, as this anyway depends on hardware specifications.

---

[5]As we utilise the publicly available C-score difficulty values, we only apply the SF to the CIFAR-10 dataset.

[6]Link to be added after acceptance

[7]Link to be added after acceptance

