# OpenReview forum: "DOES THE DEFINITION OF DIFFICULTY MATTER ? SCORING FUNCTIONS AND THEIR ROLE FOR CURRICULUM LEARNING"
_ICLR.cc/2026/Conference — Submitted to ICLR 2026_

### Official Review · Reviewer_WhVm · 2025-10-16

**Soundness:** 2
**Presentation:** 3
**Contribution:** 2
**Rating:** 2
**Confidence:** 4

**Summary:**

This paper proposes to study the role of scoring function and pacing function in curriculum learning. It aims to answer several questions such as *How do different SFs affect model performance*?

**Strengths:**

1. The topic of this paper is of great significance to the underlining mechanism of curriculum learning.

2. The paper conducts large number of experiments on the two datasets.

3. The paper is well written and easy to follow.

**Weaknesses:**

1. The conclusion of the paper is somewhat vague. On one hand, the core idea of CL is difficulty measurer and learning scheduler, thus obviously the performance of CL will be largely depended on the two components. On the other hand, the theoretical and practical role of CL in deep learning remains largely unresolved. Simply reiterating previous conclusions without offering deeper insights substantially limits the paper’s contribution.

2. For the difficulty measurer:

2.1 The authors do not explain why DSACE2020 does not exhibit similar behavior to CIFAR10; instead, they only vaguely allude to some underlying factors without concrete analysis.

2.2 In addition to ensemble-based approaches, some studies have explored dynamic hardness [1], which can also enhance model robustness. This line of work should be discussed and compared to provide a more comprehensive understanding.

3. For pacing functions: in the main paper, the authors give CL, RCL and ACL as pacing functions (Figure 3). However, only in CL, there are various pacing functions, or learning schedulers, such as predefined ones (baby step, linear, geometric, exp, etc.) and automatic ones(self-paced, transfer teacher, etc.). The discussions of pacing functions and its role to the overall performance are very limited.

4. The authors claim that anti-curriculum learning (ACL) is less preferable, and they attempt to substantiate this point in Appendix A.8. However, given that the experimental settings for curriculum learning (CL) can vary substantially, the presented evidence is not entirely convincing. As demonstrated in several experiments within the paper, conclusions derived from specific datasets cannot be assumed to generalize to others. In fact, ACL has been extensively studied in prior research [2,3], and its role and potential advantages over standard CL remain subjects of active debate.

References:

[1] Curriculum Learning by Dynamic Instance Hardness

[2] Unsupervised hard example mining from videos for improved object detection

[3] Training region-based object detectors with online hard example mining

**Questions:**

Please see weakness.

I respectfully expect deeper insights into the underlying mechanism of curriculum learning (CL) from this paper, rather than straightforward “yes” or “no” answers to the questions posed at the end of Section 1. From the current results, the main takeaway seems to be that multiple scoring functions (SFs) lack robustness across random seeds. However, this observation is rather intuitive, and employing ensemble methods appears to be the most direct remedy. I encourage the authors to further summarize and articulate the technical insights or mechanistic understanding that this paper contributes beyond empirical observations.

---

> ### Author Response · Authors · 2025-12-03
>
> We thank the reviewer for their comment. We agree that the paper needs improvement regarding its narrative, and that we must clarify the motivation and significance of our results. Furthermore, we also agree with the reviewer that the observations on our selected datasets (e.g., regarding ACL) may not be generalisable to other datasets and we should avoid making this claim, especially given the difference observed between CIFAR10 and DCASE2020, the reasons for which were unfortunately not fully resolvable from our experiments. We appreciate the hints to the publications and we will consider them for improvements of our work. Likewise, we will also aim to provide further insights beyond empirical results in future work.

---

### Official Review · Reviewer_Kah7 · 2025-10-20

**Soundness:** 3
**Presentation:** 3
**Contribution:** 2
**Rating:** 4
**Confidence:** 5

**Summary:**

This paper explores the role of scoring functions in curriculum learning, especially the robustness of scoring functions. The authors conduct a series of empirical studies to explore questions that have been less examined in prior work, including: the sensitivity of scoring functions to training settings, model architectures, and random seeds; the relationship between scoring function sensitivity and performance; and the combination of different curriculum strategies. The experiments are conducted mainly on CIFAR-10 and DCASE2020 datasets, using ResNet, EfficientNet, and PANN model architectures.

**Strengths:**

1. The topic of scoring functions is very relevant and key in the field of curriculum learning.
2. The exploration of late fusion of curriculum strategies is novel.
3. The findings and the proposed training toolkit can benefit researchers in this area.

**Weaknesses:**

1. Although the authors examine several aspects of scoring functions, the connection between these research questions is unclear. Providing a central narrative or through-line connecting these questions could make the paper more coherent.
2. Despite the number of experiments, the datasets and model architectures used are relatively small and out-dated. Consequently, the results and findings may not generalize to large-scale datasets or modern models, e.g., LLMs.
3. The proposed late fusion method is promising, but its applicability in real-world scenarios appears limited.
4. The paper confirms conclusions from prior work and concludes that curriculum learning is not a panacea, which point has already been discussed in earlier studies. Therefore, the significance of this paper is reduced.

**Questions:**

Do the authors think that the conclusions presented in this paper can be generalized to large-scale models? In the authors’ opinion, is it necessary to investigate these research question in the context of current large models?

---

> ### Author Response · Authors · 2025-12-03
>
> We thank the reviewer for their comment. We agree with reviewers’ concerns surrounding the limitations of our experiments, particularly with respect to model size and dataset size. This, however, was due to the already large experimental costs resulting from the high number of experiments we performed to explore our research questions with a reasonable robustness. Experimentation with larger models and datasets would not have been feasible with the hardware accessible for us at the moment; but we acknowledge the limitations of course. Beyond the computational costs, even though the choice of architectures may seem outdated compared to current state-of-the-art and large models, we settled on these architectures due to their establishment in literature, comparability and due to the fact that some scoring functions (especially prediction depth) had certain requirements for architecture design. Even though we can not say for certain how generalisable our results are (given also their variation across datasets), we do hope that the patterns we observe are also of considerable interest for the training of large models. We also agree with the reviewer that the narrative of the paper could be more straight-forward and potentially exclude the confirmation of already established claims from prior work – even though we do see confirmation of prior work in different contexts as an important aspect of research.

---

### Official Review · Reviewer_GHcE · 2025-10-30

**Soundness:** 2
**Presentation:** 3
**Contribution:** 1
**Rating:** 2
**Confidence:** 5

**Summary:**

The paper empirically investigates the role of sample difficulty in the performance of curriculum learning. It focuses on the empirical correlations among scores used in prior work, the reduction in variability obtained by ensembling, and the relationship between this variability and ultimate performance. The study uses two relatively small datasets: CIFAR-10 (vision) and DCASE2020 (audio). The main findings are: (i) difficulty-based rankings are sensitive to training choices, with overall moderate correlation values; (ii) somewhat higher correlation values are observed across different scores, which is possibly a new empirical observation; and (iii) this variability is not necessarily predictive of final performance improvements. The work provides additional rigorous support for empirical correlations and results reported previously.

**Strengths:**

Clarity: High, the paper is well written.

**Weaknesses:**

Originality: This is incremental work that rigorously evaluates empirical observations noted in earlier studies. In particular (as ineed noted in the paper), sensitivity to scoring functions has been observed before.
Quality: Given that the paper’s primary contribution is an empirical comparative evaluation, the scope of the evaluation is not sufficient.
Significance: In light of the points above about originality and quality, the significance of the work appears low. The hypothesis driving CL is that the ranking order produced by suitable scoring functions is crucial for CL’s success. However, this does not imply that modest correlations between rankings, when varying the training parameters, should necessarily affect final performance - as the paper itself shows. Thus, while the rigorousity of the observation is of some interest, its significance for understanding CL is limited.
Minor comment: the somewhat lengthy discussion of ERM in the upper half of page 3 (up to Eq (4)) is not necessary for an ICLR submission.

**Questions:**

Did you evaluate the correlation between SF and rate of convergence? prior work has suggested that the main contribution of CL may be speeding up convergence rather than improving ultimate performance, as this is the most robust difference between CL and random SGD traning.

---

> ### Author Response · Authors · 2025-12-03
>
> We thank the reviewer for their comment. We understand the concerns about the extent of novelty, given that some aspects closely related to our experiments have been shown in previous work already. We do, however, believe that some aspects of our study related to the stability of scoring functions and implications thereof are of crucial interest for CL. We are not certain to fully understand to which the “scope of the evaluation is not sufficient”, and we would appreciate some further insights into the type of evaluations the reviewer would deem appropriate for a similar empirical evaluation. Overall, we understand the point that the modest correlation of scoring functions themselves can not solely serve indications of final model performance, but we did highlight the importance of ordering produced by scoring functions and their overall robustness (which seems at least evident for the CIFAR experiments). We did in fact look into connections between rate of convergence and scoring functions as well within our experiments but decided against the inclusion in the paper, as they very much in line with the literature and we deemed the importance of performance more critical. However, we appreciate the feedback of the reviewer and will consider to include it in the future.

---

### Official Review · Reviewer_ZxcY · 2025-11-03

**Soundness:** 3
**Presentation:** 2
**Contribution:** 2
**Rating:** 4
**Confidence:** 4

**Summary:**

This paper provides a large-scale empirical study on curriculum learning (CL), focusing on the definition and stability of "sample difficulty". The authors evaluate a large set of scoring functions (SFs) on vision (CIFAR-10) and audio (DCASE2020) datasets. They find that a) difficulty rankings are highly sensitive to hyperparameters (model, seed, optimizer), though this can be mitigated by ensembling, b) most SFs agree on a similar notion of difficulty, and c) the use of CL does not consistently outperform standard uniform-sampling baselines. The paper's most significant finding is that models trained with different curricula (e.g., easy-to-hard vs. hard-to-easy) learn complementary knowledge, and ensembling them leads to performance improvements.

**Strengths:**

- Presents the results of  large scale experiments.
- Provides a critique to the idea that CL is a plug-and-play useful technique. Echoing prior works such as Wu et al. (2021).
- Finds that different CL schedules, such as easy-to-hard and hard-to-easy, produce models with complementary knowledge.

**Weaknesses:**

- The paper's primary conclusion that curriculum learning is not a "plug-and-play" solution is a confirmation of findings already present in the literature (e.g., Wu et al., 2021). Furthermore, the work is purely empirical, employing existing scoring functions and models without introducing novel methodology

- The paper's interesting claim that different curricula produce complementary models is supported only by indirect evidence from the performance of late-fusion ensembles. This conclusion is not convincing. The work would be significantly stronger if this claim were supported with direct evidence from interpretability analyses (e.g., feature visualization, saliency maps) or a detailed error analysis showing that CL and ACL models systematically fail on different subsets of the data. Currently, the performance gains are presented without statistical significance tests, making it difficult to understand the robustness of the ensemble effects.

- The study's conclusions are based on two mid-sized classification datasets. It is not clear how these findings would translate to other domains (e.g., NLP, RL) or to large-scale training regimes where CL is often thought to be most beneficial. Moreover, the paper notes a divergence in key results between CIFAR-10 and DCASE2020 but does not explore the reasons for this. This makes it difficult to form a generalizable conclusion.

**Questions:**

- Regarding the key difference observed in Figure 2 (robustness vs. performance correlation), what are your primary hypotheses for why this trend exists for CIFAR-10 but not DCASE2020? Could this be related to factors like dataset size, inter/intra-class variance, or the signal-to-noise ratio inherent in the different data modalities?
- Have you performed any qualitative analysis on the kinds of examples that CL-trained models correctly classify versus those that ACL-trained models handle better? This might clarify on the nature of the "learnt concepts" you mention and why they are complementary.

---

> ### Author Response · Authors · 2025-12-03
>
> We thank the reviewer for their comment and agree that the paper could be strengthened by including further analysis and experiments. We understand the concern about the fact that no new methods are being presented here, and we did not, in fact, see the conclusion of CL not being a “plug-and-play” solution as one of the key conclusions of this contribution, which we should have made clearer in the motivation and discussion. However, we certainly agree that the paper would benefit from additional analysis of the CL and ACL models including the suggested feature visualisations or qualitative evaluations, which we had not considered for this submission. We also acknowledge the limitation of the experimental evaluation in terms of the number (and corresponding diversity) of datasets, which was largely due to computational costs and the already heavy experiments to explore our research questions on the two datasets we chose. This also limits our analysis of the strong difference observed between the two datasets in Figure 2 with respect to the impact of robustness of scoring functions, as both datasets vary in many aspects from each other. All the aspects mentioned by the reviewer are clearly candidate hypotheses for being the cause of the differences, with the inter/intra-class variability or the overall complexity of the task being the strongest candidate in our opinion.

---

### Meta-Review · Area_Chair_eKDr · 2026-01-06

**Summary:**

The reviewers argue that the paper offers limited originality and significance, as its main conclusion, i.e.,  curriculum learning is not a plug-and-play solution, largely confirms prior findings and is based on a purely empirical evaluation using existing models and scoring functions. The claim that different curricula yield complementary models is weakly supported, relying only on late-fusion performance without direct evidence such as interpretability analyses, error breakdowns, or statistical significance testing. The experimental scope is narrow, using small, outdated architectures and only two mid-sized datasets, with unclear generalization to other domains or large-scale settings, and unexplained divergences between datasets (e.g., CIFAR-10 vs. DCASE2020). Key aspects of curriculum learning are insufficiently explored: 1) the role of difficulty measurers lacks analysis and comparison to related work such as dynamic hardness; 2) the discussion of pacing functions is superficial despite their central importance, and 3) claims about the inferiority of anti-curriculum learning are unconvincing given its established relevance in prior literature. Overall, the work reiterates known observations without offering deeper theoretical insight or a coherent narrative, leaving its practical impact, real-world applicability (including the proposed late-fusion approach), and contribution to understanding curriculum learning limited.

**Reviewer Concerns:**

The rebuttal is rather short and the authors have decided not to address every point raised by the reviewers in detail. The answers are rather generic and the authors are mostly asking for clarifications from the reviewers (a process suitable if the discussion would have been enabled) rather than attempting to provide exhaustive explanations to the questions of the reviewers.

**Reviewer Scores:**

I think the discussion might have improved some of the opinions of the reviewers but I doubt the overall outcome would have been different. The main problem is that all the initial scores were rather negative and the concerns were real. This has been partially acknowledged by the authors in their rebuttal even if the details are rather scarse.  Based on this, I have serious doubts that the reviewers (at least the very negative ones) would have updated their scores.  Overall, I think the authors should have allocated more time for their rebuttal in order to have any chance to obtain a change in the final decision.

---

### Decision · Program_Chairs · 2026-01-26

Reject